# Designing of Drug Delivery Systems to Improve the Antimicrobial Efficacy in the Periodontal Pocket Based on Biodegradable Polyesters

**DOI:** 10.3390/ijms25010503

**Published:** 2023-12-29

**Authors:** Magdalena Zięba, Wanda Sikorska, Marta Musioł, Henryk Janeczek, Jakub Włodarczyk, Małgorzata Pastusiak, Abhishek Gupta, Iza Radecka, Mattia Parati, Grzegorz Tylko, Marek Kowalczuk, Grażyna Adamus

**Affiliations:** 1Centre of Polymer and Carbon Materials, Polish Academy of Sciences, 34. M. Curie-Skłodowska St., 41-819 Zabrze, Poland; mzieba@cmpw-pan.edu.pl (M.Z.); wsikorska@cmpw-pan.pl (W.S.); mmusiol@cmpw-pan.pl (M.M.); hjaneczek@cmpw-pan.pl (H.J.); jwlodarczyk@cmpw-pan.pl (J.W.); mpastusiak@cmpw-pan.pl (M.P.); mkowalczuk@cmpw-pan.pl (M.K.); 2Department of Optoelectronics, Silesian University of Technology, B. Krzywoustego 2, 44-100 Gliwice, Poland; 3Faculty of Science and Engineering, School of Pharmacy, University of Wolverhampton, Wulfruna Street, Wolverhampton WV1 1LY, UK; a.gupta@wlv.ac.uk; 4Faculty of Science and Engineering, Wolverhampton School of Life Sciences, University of Wolverhampton, Wulfruna Street, Wolverhampton WV1 1LY, UK; i.radecka@wlv.ac.uk (I.R.); m.parati@wlv.ac.uk (M.P.); 5Department of Cell Biology and Imaging, Institute of Zoology and Biomedical Research, Faculty of Biology, Jagiellonian University, Gronostajowa 9, 30-387 Kraków, Poland; grzegorz.tylko@uj.edu.pl

**Keywords:** poly(L-lactide-*co*-glycolide)/poly[(R,S)-3-hydroxybutyrate], electrospun nonwoven delivery system, biodegradable polyesters, antimicrobial, proanthocyanidins, periodontal

## Abstract

Delivery systems for biologically active substances such as proanthocyanidins (PCANs), produced in the form of electrospun nonwoven through the electrospinning method, were designed using a polymeric blend of poly(L-lactide-co-glycolide) (PLGA)and poly[(R,S)-3-hydroxybutyrate] ((R,S)-PHB). The studies involved the structural and thermal characteristics of the developed electrospun three-dimensional fibre matrices unloaded and loaded with PCANs. In the next step, the hydrolytic degradation tests of these systems were performed. The release profile of PCANs from the electrospun nonwoven was determined with the aid of UV–VIS spectroscopy. Approximately 30% of the PCANs were released from the tested electrospun nonwoven during the initial 15–20 days of incubation. The chemical structure of water-soluble oligomers that were formed after the hydrolytic degradation of the developed delivery system was identified through electrospray ionization mass spectrometry. Oligomers of lactic acid and OLAGA oligocopolyester, as well as oligo-3-hydroxybutyrate terminated with hydroxyl and carboxyl end groups, were recognized as degradation products released into the water during the incubation time. It was also demonstrated that variations in the degradation rate of individual mat components influenced the degradation pattern and the number of formed oligomers. The obtained results suggest that the incorporation of proanthocyanidins into the system slowed down the hydrolytic degradation process of the poly(L-lactide-*co*-glycolide)/poly[(R,S)-3-hydroxybutyrate] three-dimensional fibre matrix. In addition, in vitro cytotoxicity and antimicrobial studies advocate the use of PCANs for biomedical applications with promising antimicrobial activity.

## 1. Introduction

Drug delivery devices (DDSs) are vehicles for the transportation of therapeutic agents; as well, they can maximize the therapeutic effects of the delivered drug while minimizing undesired side effects [1,2]. Conventional drug delivery methods are the most popular route for drug administration because they have advantages such as ease of use and a very high degree of dosage flexibility. However, systemic administration pathways also have disadvantages, such as serious side effects, adverse biodistribution, low selectivity of therapeutic effect, burst release of the drug, and damage to healthy cells [3,4]. DDSs in the oral cavity can prepared via various methods to adapt to complex disease situations. Carriers’ materials used in recent years as oral drug delivery systems can be prepared via various methods to adapt to individual patient needs. DDSs which can be used in the oral cavity have been classified as nanoparticles, hydrogels, nanofibres, and films [5,6,7]. Among all these options, electrospinning polymeric fibres (EPFs) produced with biodegradable and biocompatible polymers have proven to be an interesting strategy for drug delivery system applications [5,6,7].

Periodontitis is an extremely common chronic inflammatory disease characterized by periodontal tissue destruction, usually caused by the accumulation of pathogenic microorganisms [2,8]. According to a World Health Organization report, it is one of the world’s most widespread chronic ailments occurring after the age of 35. Recently, there has been growing interest in controlled drug delivery systems in oral infectious disease as a potential method to address all these challenges [9]. Surgical intervention, mechanical therapy, and the use of pharmacological agents are among the most popular approaches used for the treatment of periodontitis [10,11]. It is worth mentioning that sustained local administration of drugs into the periodontal pockets in the mouth is problematic due to the continuous secretion of saliva, which makes it hard for the administered drug to remain in the periodontal pocket, resulting in being swallowed by the patient [12]. Drug delivery devices improve antimicrobial efficacy in the periodontal pocket and demonstrate other clinical benefits.

An important step in the preparation of such delivery systems is the selection of appropriate polymer matrix, which should be non-toxic to mammalian cells, biodegradable, and flexible enough to suit the wound topography well [2,13].

The aliphatic polyesters of alpha- and beta-hydroxy acids (polylactide and its copolymers with glycolide, caprolactone, and polyhydroxybutyrate) are of great importance in this field [14]. The drugs dispersed in such matrices are released both via diffusion and via erosion of the carrier. Poly(lactic-co-glycolic acid) copolymers consisting of less than 30% glycolide units are the most successfully used synthetic biodegradable polymers in the medical field. Biocompatibility, biodegradability, flexibility, and minimal side effects are the main advantages of using this polymer for such an application [15]. Recently, poly(L-lactide-co-glycolide) (PLGA) biopolyesters in the form of implants [16], disks [17] or dental films [18] have been used in the treatment of periodontium for better local administration of antibiotics and for reducing their side effects.

Proanthocyanidins (PCANs) are oligomers or polymers of monomeric flavan-3-ols produced as an end-product of the flavonoid biosynthetic pathway [19,20,21]. PCANs exhibit antioxidant, anti-inflammatory, cytoprotective, and antimicrobial activity. Proanthocyanidins are a unique class of phytonutrients for the prevention and treatment of periodontal diseases due to their high antimicrobial and immunomodulatory activities. PCANs can also be beneficial in adhesive dentistry due to their natural dentin cross-linking activity and inhibition of MMPs [20,21]. Using natural, non-toxic products can make a significantly effective product in alleviating periodontitis symptoms and preventing the disease from progressing [21]. At the moment, there is growing interest in incorporating bioactive substances into polymer matrices, for example in the form of electrospun mats. The simple and powerful technique of electrospinning involves preparation of polymeric fibres with different diameters, from a few microns to tens of nanometres. Nanofibres can be fabricated from natural or synthetic polymers and blended with various additives. Due to control of infection and small pores, electrospun mats are very beneficial for preventing the penetration of bacteria [22,23,24,25]. Periodontal treatment involves the preparation of many oral nanofibrous mats that have antibacterial, anti-inflammatory, and tissue regeneration properties [5,26]. Electrospun nanofibres are capable of mimicking the natural extracellular matrix (ECM) while allowing for the promotion of cell adhesion, proliferation, differentiation, and migration Also, nanofibres can direct cell growth, and they favour precisely oriented tissues, such as stratified periodontal fibres. Hence, electrospun drug-loaded nanofibres have the potential to be used effectively in periodontal treatment [5,26].

This study aimed to develop a polymeric drug delivery system intended for potential use in periodontitis treatment with the use of proanthocyanidins extracted from *Pelargonium sidoides* as an active substance. *Pelargonium sidoides* (PSRE) is a medicinal plant originating from coastal regions of southern Africa and is highly valued by the native population for its curative properties. The antibacterial effect of this plant is mainly attributed to the proanthocyanidins it contains, which also have antioxidant, anti-inflammatory, anti-aging, and anti-cancer properties [27,28].

The strategic aspect of this research is related to the methods of incorporation of bioactive substances into polymeric carriers.

Our previous research demonstrated the possibility of obtaining polymer delivery systems for potential wound treatment application in the form of electrospun mats with flexible properties ensuring compatibility with the wound topology from the poly(D,L)lactide/poly[(R,S)-3-hydroxybutyrate], P(D,L)LA/(R,S)-PHB, polymeric blend containing PCANs [29].

In this study, we focused on designing drug delivery devices to improve antimicrobial efficacy in the periodontal pocket based on biodegradable polyesters. As a polymer carrier, a blend of bioresorbable polyesters poly(L-lactide-co-glycolide) with synthetic poly[(R,S)-3-hydroxybutyrate], (PLGA/(R,S)-PHB) in the form of electrospun mats was used. Our intention was to develop a system that would enable the controlled release of biologically active PCANs over the defined period during therapy without the need to remove the system. Due to the hydrolytic degradation of the developed electrospun nonwoven fabrics over time, the resulting degradation products should be gradually resorbed by the body and completely absorbed after therapy. 

These studies included the preparation of polymer blends of poly(L-lactide-*co*-glycolide) copolymer, PLGA with synthetic poly[(R,S)-3-hydroxybutyrate], (R,S)-PHB with a chemical composition of PLGA/(R,S)-PHB of 80/20 wt%, and examined the possibility of using the developed biomaterial for the production of delivery systems in the form of electrospun mats with the desired properties, enabling the loading of the biologically active substance, i.e., PCANs. In the next stage, the structural and thermal characterization of the obtained PLGA/(R,S)-PHB electrospun nonwoven were investigated, and the comparative hydrolytic degradation tests of the developed systems, both with and without loaded PCANs, were carried out. The results of the hydrolytic degradation studies allowed us to determine the PCAN release profile from the elaborated system and determine the rate of the electrospun nonwoven hydrolysis dependent on PCAN presence. Additionally, structural characterization of the degradation products of the developed polymeric carrier was performed. The cytocompatibility and antimicrobial activity of the developed system were also tested.

## 2. Results and Discussion

The delivery system of PCANs was designed from a biodegradable polymeric blend in the form of a three-dimensional fibre matrix. The electrospun nonwovens studied were prepared from a polymeric blend of poly(L-lactide-co-glycolide) and poly[(R,S)-3-hydroxybutyrate] (PLGA/(R,S)-PHB; 80 wt%/20 wt%) using electrospinning. The electrospinning process typically produces randomly oriented, electrospun nonwoven. Our previous studies on the polymer solution concentration showed that increase of polymer concentration causes a slower sedimentation of bioactive compound particles in syringe during electrospinning process. Very good fibres were produced using a 10% *w*/*v* polymeric blend solution in HFIP [29]. Thus, this concentration was used in the production of electrospun mats from PLGA/(R,S)-PHB blend with and without proanthocyanidins. The other solution properties (density, viscosity, surface tension, etc.) and operating parameters (flow rate, electric field strength, and electric current flow) presented in the section Electrospinning Experiment were also significant parameters to obtain the desired proper size of fibre diameter.

Figure 1 and Figure 2, below, show the ^1^H-NMR spectrum and DSC traces of the resulting PLGA/(R,S)-PHB electrospun nonwovens.

The ^1^HNMR spectrum revealed the presence of signals corresponding to protons of repeating lactidyl (LA), glycidyl (GA) and 3-hydroxybutyrate (3-HB) units, characteristic of the mixture components (PLGA and poly[(R,S)-3-hydroxybutyrate]) according to structure (Figure 1). The chemical composition of electrospun nonwoven samples before degradation was calculated based on the integral values of signals 4 at 1.28 ppm corresponding to the protons of the methyl group 3-HB of the poly[(R,S)-3-hydroxybutyrate] component and signals 1 at 1.58 ppm and 3 at in the range of 4.5–5 ppm represent the protons of the methyl groups LA and the methylene group GA of the PLGA copolymer, respectively. The molar ratio of LA, GA and 3-HB units in the electrospun nonwoven fabric determined on the basis of the 1H-NMR spectrum was 60, 25 and 15 mol%, respectively.

In Figure 1, signals from PCANs were not observed because PCANs do not dissolve in CDCl_3_.

The second DSC heating traces for PLGA/(R,S)-PHB three-dimensional fibre matrix without PCANs show only one glass transition temperature, T_g_, equal to 38.9 °C, which is located between the T_g_ values of the blend components (see Figure 2). Thus, thermal analysis confirms that the electrospinning process does not disturb the miscibility of the blend components. In the case of the electrospun nonwoven loaded with PCANs in DSC trace, two T_g_ values were observed, which indicates that the addition of PCANs to the PLGA/(R,S)-PHB polymer system disrupted its compatibility.

### 2.1. Degradation of PLGA/(R,S)-PHB and PLGA/(R,S)-PHB/PCAN Electrospun Nonwoven

The polymeric electrospun nonwovens were subjected to hydrolytic degradation. The progress of the mats’ hydrolysis was assessed via macro- and microscopic observations as well as molar mass, structure, and thermal properties changes of samples studied. The visual inspection of initial PLGA/(R,S)-PHB and PLGA/(R,S)-PHB/PCAN nonwoven fabric as well as their changes during incubation was carried out with aid of POM microscopy, as shown in Figure 3.

The digital images, as well as the POM micrographs, show that the surface of PLGA/(R,S)-PHB mats before incubation was smooth, and the bioactive substance was distributed in the polymer matrix (Figure 3). After degradation, pitting and cracks were observed on the surface of both types of mats, which are layered in nature. It is especially visible in the case of a mat loaded with PCANs. The morphology of the PLGA/(R,S)-PHB three-dimensional fibre matrix were also investigated with aid of SEM microscopy, and the results are shown in Figure 4.

As shown in the SEM micrographs, the bioactive substance formed different-sized agglomerates, which are irregularly intercalated between the fibres (Figure 4). This is probably caused by the very poor solubility of the bioactive substance in organic solvents and the impossibility of keeping a solution homogenous when it is used in the electrospinning process. The fibres in PLGA/(R,S)-PHB electrospun nonwoven studied before degradation possessed average diameters of 2.2 µm. The addition of PCANs caused disorganization and entanglement of fibres and influenced varying diameters of single fibres. The average diameters of the fibres after loading of PCANs were in the range of 2.0–3.0 μm. Disordered fibres in the electrospun nonwoven have an impact on the formation of porous structures with different densities [30,31]. The appearance of the sample’s surface morphology under the higher magnification resulted from the gradual softening of the material during the analysis.

The changes in number–average molar mass of the samples during the degradation process were monitored via gel permeation chromatography (GPC). All investigated samples show the systematic shifting of GPC traces to a lower molar mass value with the progress of the degradation process (Figure 5).

The results of this measurement revealed that M_n_ values for mats both without and with PCANs decreased and reached a value at the level of 1000 g mol^−1^ after 71 days of incubation. The overlapping of the GPC curves may indicate differences in the degradation profile of the polymeric mat components. It is worth noting that the molar mass loss of the PLGA/(R,S)-PHB sample with PCANs is lower than for the PLGA/(R,S)-PHB mats not loaded with a biologically active substance. This indicates that the addition of PCANs has an influence on the behaviour of three-dimensional fibre matrix of PLGA/(R,S)-PHB during degradation.

The ^1^H-NMR spectra of PLGA/(R,S)-PHB electrospun nonwoven without and with a biologically active substance, i.e., PCANs, before and after degradation time are shown in Figure 6.

The resulting ^1^H-NMR spectra show the signals of protons characteristic of the LA, GA, and 3-HB repeating units of the polymeric components of electrospun nonwoven accordingly to the structure presented in Figure 1. In addition, in the spectra obtained after 71 days of degradation, proton signals were detected that were characteristic of low-molecular-mass 3-hydroxybutyrate (OHB), and OLAGA cooligoesters terminated with hydroxyl and carboxyl end groups. These oligomers are formed during hydrolyses of the electrospun nonwoven components (R,S)-PHB and PLGA, respectively [32]. The presence of OHB and OLAGA (co)oligomers proton signals with a simultaneous decrease of the intensity of the signals characteristic for repeating units LA, GA, and 3-HB of polymers indicates the progress of the hydrolysis process. Random cleavage of the polymer chains leads to a decrease in the molar mass of the components of the blend. This was confirmed by the GPC results (see Figure 5). It should also be noted that no signals from PCANs were observed in the obtained ^1^H-NMR spectra. This is because PCANs do not dissolve in CDCl_3_.

Moreover, on the basis of ^1^H-NMR spectra recorded after a certain period of degradation, changes in the chemical composition of electrospun nonwovens PLGA/(R,S)-PHB and PLGA/(R,S)-PHB/PCAN were also estimated, and the results are presented in Figure 7.

The (R,S)-PHB component content in three-dimensional fibre matrix samples remaining after a specific incubation period was calculated based on the integral value of signals corresponding to the protons of the methyl groups of both 3-HB repeating units and OHB oligomers formed during degradation [32].

Figure 7 shows that the content of the PLGA component in the tested mats without and with PCANs decreases, while the content of the PHB component increases. However, it was observed that changes in the chemical composition of the PLGA/(R,S)-PHB electrospun nonwoven started from the beginning of the incubation process, while in the case of the PCAN-loaded mat, a decrease in PLGA content was observed after 51 days of degradation. The lower degradation rate of the PLGA component in this case is probably related to the presence of PCANs, which interferes with the penetration of water into the electrospun nonwoven.

Figure 8 presents the second DSC heating traces for PLGA/(R,S)-PHB and PLGA/(R,S)-PHB/PCAN electrospun nonwoven samples remaining after 71 days of hydrolytic degradation.

A decrease in T_g_ from 38.9 ° C to 17.9 °C was observed for the sample without PCANs after 71 days of incubation. In contrast, for the sample with PCANs, a decrease of T_g_ value from 47 °C to 28.9 °C was observed (see Figure 2 and Figure 8). The observed differences in the thermal properties of the tested mats probably resulted from the different rates of hydrolytic degradation of the loaded and unloaded PCAN mats, leading to the formation in the degradation process of a different amount of oligomers, which remained in the tested mats. The PLGA/(R,S)-PHB electrospun nonwoven degraded faster, and the oligomers formed acted as plasticizers of the system. In the case of PLGA/(R,S)-PHB/PCAN electrospun nonwoven loaded with PCANs, the biologically active substance located between the fibres of the polymer matrix probably hinders the absorption of water, which slows down the hydrolysis process, and the observed amount of low-molar-mass oligomeric degradation products formed was lower. The presence of PCANs between the fibres inside the electrospun nonwoven was confirmed with SEM analysis (see Figure 4). The progress of the hydrolytic degradation of PLGA/(R,S)-PHB and PLGA/(R,S)-PHB/PCAN three-dimensional fibre matrix was also monitored via mass loss measurements.

Figure 9 shows the mass loss for the PLGA/(R,S)-PHB and PLGA/(R,S)-PHB/PCAN electrospun nonwoven observed during degradation carried out in water.

The observed mass loss of the tested samples during their incubation in water is related to the release of low-molar-mass degradation products of the mats’ polymer components into the water environment, as well as, in the case of the PLGA/(R,S)-PHB/PCAN electrospun nonwoven, by the release of the bioactive substance contained therein. At the first stage of incubation, up to 51 days, a higher mass loss was observed in the PLGA/(R,S)-PHB/PCAN electrospun mat samples than in the PLGA/(R,S)-PHB ones. A significant increase in the mass loss of PLGA/(R,S)-PHB/PCAN sample after 51 days of incubation can be connected to the fact that the three-dimensional fibre matrix starts to disintegrate after this incubation time, and the release of PCANs is facilitated. The observed violent increase in mass loss after 51 days of incubation may additionally arise from the increased rate of degradation of the matrices themselves, which is facilitated by the smaller quantity of PCANs contained therein. The above observations were confirmed by the results obtained from the ^1^H-NMR analysis. During the first stage, the differences in mass loss of the electrospun nonwoven samples resulted mainly from the release of a biologically active substance, i.e., PCANs. During the second stage, starting on day 51, due to the increased migration of low-molar-mass degradation products from the mats into the water, in both electrospun nonwoven samples, greater mass loss was observed. Meanwhile, in the case of PLGA/(R,S)-PHB electrospun nonwoven, mass loss was systematically higher than in the PLGA/(R,S)-PHB/PCAN samples; after 71 days of degradation, mass losses of 51.1% and 38.5% were observed for the PLGA/(R,S)-PHB and PLGA/(R,S)-PHB/PCAN samples, respectively. Thus, PCANs in the electrospun nonwoven slow down degradation, which may be due to the presence of polymer–PCAN physical interactions such as ion–ion attraction/repulsion, hydrogen bonding, and van der Waals forces, which can significantly alter the degradation time and hinder the release of oligomeric degradation products from the three-dimensional fibre matrix into the degradation environment.

### 2.2. Release Study of PCANs from PLGA/(R,S)-PHB and PLGA/(R,S)-PHB/PCAN Electrospun Nonwoven 

Figure 10 shows the release profile of PCANs from PLGA/(R,S)-PHB/PCAN electrospun nonwoven during incubation in water.

Comparing the mass loss of the tested mat and the quantity of released PCANs from the beginning of incubation, the PCAN release process is accompanied by the degradation of the polymer mat and the release of polymeric degradation products into the degradation medium. At the beginning of the incubation, a rapid release of PCANs from the mat occurred up to the level of 20 wt%, then from the 15^th^ day, a gradual, more moderate release of PCANs reached about 50% after 51 days of incubation. The observed further increase in the quantity of PCANs released after 51 days is because the three-dimensional fibre matrix starts to disintegrate after this incubation time. The increased release of low-molar-mass oligomeric degradation products from the samples into the degradation environment (Figure 9) is accompanied by a facilitated release of the PCANs occluded inside the fibre matrix.

Depending on the type of preparation polymer and bioactive compound, sample procedure, and degradation conditions, the kinetics of release may be influenced by one or more physical or chemical phenomena [33]. In this case, the biologically active substance is released via diffusion from the polymer matrix into the degradation medium. With the progressive release of PCANs and microenvironmental changes in pH inside the matrix caused by OHB, OLA, and OGA oligomers formed, the degradation process of the PLGA/(R,S)-PHB/PCAN fibre matrix should be less disturbed over time, however, the degradation of this samples was still slower than the electrospun nonwoven without PCANs.

### 2.3. ESI-MS Study of the Degradation Products Released from PLGA/(R,S)-PHB and PLGA/(R,S)-PHB/PCAN Electrospun Nonwoven 

The chemical structure of the low-molar-mass oligomeric degradation products which were released from the PLGA/(R,S)-PHB and PLGA/(R,S)-PHB/PCAN three-dimensional fibre matrix into the degradation medium was determined with the aid of ESI-mass spectrometry. The ESI-mass spectra recorded for the water solutions collected after 71 days of incubation of PLGA/(R,S)-PHB and PLGA/(R,S)-PHB/PCAN samples are shown in Figure 11a,b, respectively.

The oligomers of the lactic acid OLA and OLAGA copolyester oligomers as well the oligomers of poly(3-hydroxybutyrate), OHB terminated with hydroxyl and carboxyl end groups were identified as low-molar-mass degradation products released into water during the incubation of the polymeric mats studied. The ESI-mass spectrum recorded for the water solution collected after 71 days of incubation of PLGA/(R,S)-PHB sample, shown in Figure 11a, is less complicated. One maximum can be distinguished in it with the visible major series of ions at *m*/*z* 557, 643, 729, 815, 901, and 987 (with mass increment between the signals equal to 86 Da). Those signals represent sodium adducts of 3-HB oligomers terminated with hydroxyl and carboxyl end groups. The structure of oligomers visible on ESI-mass spectra was confirmed with aid of ESI-MS^n^ experiments. The conducted tests confirmed that the PLGA matrix component degraded faster than the PHB component. During the incubation, PLGA copolyester systematically degraded to oligomers with lower and lower molar mass and, consequently, lactic and glycolic acids. Therefore, 3-HB oligomers dominated the solutions after an increasingly longer periods of degradation. The two major maxima of singly charged positive ions were observed on the ESI-mass spectrum recorded for the solution collected after incubation of the PLGA/(R,S)-PHB/PCAN sample and presented in Figure 11b. The main series of ions at *m*/*z* 257, 328, 401, 473, 545, and 617 (with a mass spacing of 72 Da) located in the area of the first maximum correspond to the sodium adducts of lactic acid oligomers with hydroxyl and carboxyl end groups. Moreover, in the range of the first maximum, the ions representing sodium adducts of copolyester oligomers with hydroxyl and carboxyl end groups can also be noticed. At the second maximum, located in the mass range of *m*/*z* 600–1000, the visible major series of ions at *m*/*z* 557, 643, 729, 815, 901, and 987 (with the mass increment between the signals equal to 86 Da) corresponds to sodium adducts of 3-HB oligomers terminated with hydroxyl and carboxyl end groups.

It is worth noting that the mass spectrometry analysis confirmed that the fibre matrix containing PCANs decomposed more slowly than the samples without the addition of this substance. The slower degradation of the three-dimensional fibre matrix is probably due to the interactions between the PCANs found on the fibres of the electrospun nonwoven, the polymer components, and the resulting degradation products. Limited access to water in the interior of the electospun nonwoven caused by the presence of PCANs slowed down the hydrolysis process and hindered the release of oligomeric degradation products into the degradation environment.

### 2.4. Cytocompatibility Test (MTT Assay)

Cytocompatibility stands as a fundamental attribute of a material when it comes to its utilization in the biomedical sector. In this current research, an in vitro MTT assay was employed to assess the cytocompatibility of PLGA/(R,S)-PHB electrospun nonwoven, both without and with the 20 wt% addition of the biologically active substance, i.e., PCANs. The MTT assay results indicated varying levels of cytocompatibility of PLGA/(R,S)-PHB with the selected cell lines: U251MG, MSTO, and PANC 1. MSTO demonstrated the highest cell viability (90.91 ± 6.54%) when exposed to conditioned DMEM media, followed by PANC 1 and U251MG (Table 1). These results confirmed that the U251MG cell line is sensitive to PLGA/(R,S)-PHB. 

When cell lines were exposed (in vitro) to DMEM conditioned with PLGA/(R,S)-PHB/PCAN, the viability was significantly reduced (*p* < 0.05) for all three cell lines. MSTO cell line demonstrated highest cell viability (%) out of all the three tested cell lines (Table 1). These findings are in accordance with our previous study where the cytotoxicity of P(D,L)LA/(R,S)-PHB with 20 wt% PCANs was reported [29]. When PCANs containing electrospun nonwoven were produced with PLGA/(R,S)-PHB, the viability of MSTO cell lines was recorded to be better than P(D,L)LA/(R,S)-PHB, with 20 wt% PCANs. These findings confirm that PCANs at this concentration (20 wt%) demonstrate the cytotoxic effect in the in vitro test settings, but different cell lines respond differently, with MSTO being the most viable vs. PANC 1 being least (Table 1). 

PCANs possess an established antioxidative characteristic that affects numerous signalling pathways including, nuclear factor erythroid 2-related factor 2 (Nrf2), mitogen-activated protein kinase (MAPK), nuclear factor-kB (NF-κB), and phosphoinositide 3-kinase/protein kinase B (PI3K/Akt) [34]. The decrease in cell viability, as shown in Table 1 and Figure 12, might be linked to the free radical scavenging capacity of PCANs, which can impact signalling pathways [35]. Nonetheless, it is important to note that PCANs have a demonstrated safety profile, advocating for its safe application in clinical medicine [34]. Furthermore, PCANs have a strong antimicrobial property, advocating for the application of these electrospun nonwovens in periodontal pockets [29]. However, further work is required to evaluate the behaviour (in vivo) of these electrospun nonwovens before their clinical application as a drug delivery system for periodontal application to control infections.

### 2.5. Antibacterial Activity (Disc Diffusion Assay, DDA) 

Periodontitis is a chronic infection that affects the supporting tissues of the teeth. This can lead to gingival inflammation, gingival attachment loss, and alveolar bone resorption and can lead to tooth loss [36]. One-stage full-mouth disinfection is one of the non-invasive therapeutic strategies employed to treat periodontitis [28]. Attributed to antioxidative, anti-inflammatory and antimicrobial activities, proanthocyanidins (PCANs) have attracted a wide research interest for their biomedical applications including in the periodontal treatment [28,37]. Our group has previously reported the antimicrobial properties of PCANs from *Pelargonium sidoides* [29]. In the current study, the work was extended by testing the antimicrobial properties of varied concentrations of PCANs, ranging from 2.5% to 20% *w*/*v*. The test (in vitro) results revealed that PCANs at a 2.5% *w*/*v* concentration is capable of inhibiting Gram-positive *Staphylococcus aureus*. At higher concentrations, the antimicrobial activity increased (Figure 13), with 20% exhibiting the highest antimicrobial activity. These results support the application of PCANs to control infections in periodontal disease.

## 3. Materials and Methods

### 3.1. Materials

Poly(L-lactide-*co*-glycolide), PLGA was obtained via ring-opening polymerization of L-lactide and glycolide (both monomers HUIZHOU Foryou Medical Devices Co., Ltd., Huizhou, China) in the presence of biocompatible zirconium (IV) acetylacetonateZr(Acac)_4_ initiator (Sigma Aldrich, Merck KGaA, Darmstadt, Germany). The molar ratio of Zr(Acac)_4_ to the sum of moles of monomers was equal to 1:600. The process was performed in bulk at 130 °C for 24 h and then at 115 °C for 72 h in an argon atmosphere. Unreacted monomers were removed from the bulk via dissolution in chloroform and precipitation in methanol (both Sigma Aldrich, Merck KGaA, Germany). Finally, the purified material was dried under vacuum (20 mb; 20 °C) to constant weight [38]. The PLGA copolymer with M_w_ of 104,000 gmol^−1^ and dispersity 2.48 was obtained. The molar ratio of L-lactide to glycolide units in the copolymer was 70 to 30 mol% as determined from ^1^H-NMR spectra. The glass transition temperature, T_g_, determined via DSC, was 58 °C. Poly[(R,S)-3-hydroxybutyrate], (R,S)-PHB was synthesized via bulk polymerization of (R,S)-β-butyrolactone, using tetrabutylammonium acetate as the initiator. The (R,S)-PHB with M_n_ of 1000 g mol^−1^ and dispersity 1.2 was used [39]. A poly(L-actide-*co*-glycolide) and poly[(R,S)-3-hydroxybutyrate] blend with weight ratio equal 80 wt% to 20 wt% was prepared by mixing appropriate amounts of PLGA and (R,S)-PHB in hexafluoroisopropanol (HFIP, Sigma-Aldrich Chemie GmbH, Steinheim, Germany), directly before the electrospinning experiment.

The U251MG glioblastoma cell line was obtained from (ATCC, Glasgow, UK), along with the MSTO mesothelioma and PANC 1 pancreatic ductal adenocarcinoma cell lines. Ringer solution (1/4 strength) tablets were purchased from (Lab M, Heywood, UK) and prepared according to the manufacturer’s protocol. Thiazolyl blue tetrazolium bromide (MTT) and sodium bicarbonate were acquired from Sigma-Aldrich, Gillingham, UK). For the preparation of Sorensen’s glycine buffer, NaCl (5.8 g/L) and glycine (7.6 g/L) were used and sourced from Sigma-Aldrich, (UK). Dimethyl sulphoxide (DMSO) was purchased from (Alpha Aesar, Lancashire Heysham, UK). Trypsin was purchased from Lonza (Belgium). Dulbecco’s Modified Eagle’s Medium (DMEM), fetal bovine serum (FBS), L-glutamine, and antibiotic Antimycotic (comprising 10,000 units/mL penicillin, 10,000 µg/mL streptomycin and 25 µg/mL amphotericin B) were obtained from Gibco (London, UK). 

*Staphylococcus aureus* (NCIMB 6571) was obtained from the University of Wolverhampton culture collection and maintained at −20 °C in a lyophilised form. Stock culture was resuscitated on sterile tryptone soy agar (TSA) and sterilised via autoclaving (Priorclave, London, UK) prior to use.

### 3.2. Electrospinning Experiment

The electrospinning method (TL-Pro-BM electrospinning unit, TongLi Tech, Shenzen, China) was used for the preparation of the PLGA/(R,S)-PHB three-dimensional fibre matrix, with and without proanthocyanidins (PCANs). The concentration of PLGA/(R,S)-PHB solution for the electrospinning was 10% *w*/*v* in HFIP. To prepare the suspension for electrospinning PCANs, 20% in weight ratio to the blend was ground in a mortar and then added as a powder to the polymer solution. A microfibrous mat was formed under the following process parameters. The difference in voltage potentials between the spinning nozzle (steel needle G22) and the fibre collector (a steel mandrel with a diameter of 27 mm, rotating at a speed of 400 RPM) was 30 kV, and the distance between mentioned parts of the electrospinning unit was 23 cm. The suspension was dosed to the spinning nozzle via syringe pump (PHD Ultra 4400, Harvard Apparatus, Holliston, MA, USA) with a flow rate of 3 mL per hour. To limit the sedimentation of PCANs, vertical setting of the tubeless spinneret was applied. Also, the suspension was stirred inside the container with a magnetic stirrer. Electrospinning was carried out at 20 °C, at relative humidity equal to 30%.

### 3.3. Hydrolytic Degradation under Laboratory Condition

For the degradation experiments, three-dimensional fibre matrix of PLGA/(R,S)-PHB, with and without proanthocyanidins were incubated at 37 °C (± 0.5 °C) in screw-capped vials with an air-tight PTFE/silicone septum containing 10 mL of demineralized water (pH = 6.1) The solid samples of electrospun mats were withdrawn from the test environment after specific time intervals, specifically 1, 3, 5, 7, 9, 11, 15, 19, 23, 27, 31, 35, 39, 43, 47, 51, 55, 65, 71, and 85 days. The incubation of the samples was run in triplicate. After a predetermined degradation time, the samples were separated from the degradation medium, washed with demineralized water and dried, first on filter paper and then under vacuum at a temperature of 25 °C to a constant mass. The changes in the molar mass, structure, thermal properties, and mass as well as the visual and microscopic examination of surface of mats before and after degradation, were determined. The percentage mass loss of electrospun nonwoven after degradation, Δm_d_, was evaluated by comparing the dry mass (md) at the specific incubation time intervals with the initial mass (m_0_). 

Additionally, degradation media were collected, and then analysed by means of ESI-MSin negative ion mode to evaluate the progress of hydrolytic degradation and via UV–VIS to evaluate the release of the bioactive substance (PCANs) from electrospun nonwoven studied.

### 3.4. Biological Studies

#### 3.4.1. Cytocompatibility Study (In Vitro MTT Assay) 

In the current study, the effect of the three-dimensional fibre matrix (without and with PCANs) on cell viability was investigated in vitro using the mammalian cancer cell lines U251MG (human glioblastoma), MSTO (human mesothelioma), and PANC1 (human pancreatic ductal adenocarcinoma) following the similar protocol previously reported by our group [29]. Each cell line was cultured in Dulbecco’s Modified Eagle Medium (DMEM medium) containing 4.5 g/L glucose, supplemented with fetal bovine serum (10%), antibiotic Antimycotic (1%), L-glutamine (2 mM) and incubated at 37 °C in a humidity incubator with 5% CO_2_. Briefly, 25,000 cells per well were seeded in 24-well plates for 24 h at 37 °C in a 5% CO_2_ incubator. The medium was conditioned with electrospun mats of PLGA/(R,S)-PHB or PLGA/(R,S)-PHB/20% PCANs (1 cm^2^ electrospun mats in 10 mL DMEM) overnight under agitated conditions at 4 °C. The cells were then exposed to this medium conditioned with PLGA/(R,S)-PHB or PLGA/(R,S)-PHB/20% PCANs. The in vitro cytotoxic impact of the samples on the selected cell lines was assessed using the standard MTT cytotoxicity assay. This involved adding a 5 mg/mL MTT solution (obtained from Sigma, UK) to all the wells, incubating for 2 h, and then dissolving the formazan crystals using a mixture of DMSO and Sorensen’s glycine buffer at pH 10.5. Following a 24 h incubation period, the morphology of cells, confluence of the cell monolayer, and cell viability were observed microscopically using an inverted light microscope (Nikon, Tokyo, Japan) and representative optical photomicrographs of cells were captured at 10× magnification. Cell viability was calculated using the mean (n = 4) absorbance measured and the results were statistically analysed via ANOVA with a Tukey’s multi comparisons test using GraphPad Prism (version 7.02). 

#### 3.4.2. Antibacterial Activity (Disc Diffusion Assay, DDA)

The antimicrobial activity of PCANs extract obtained from *Pelargonium sidoides* was investigated against *S. aureus* using the disc diffusion assay. Sterile inert cellulosic discs (8 mm) were aseptically loaded with varied concentrations (2.5%, 5%, 10%, and 20% *w*/*v*) of aqueous PCANs under constant agitated conditions (180 rpm) in an orbital shaker overnight (Innova^®^ 43, Santa Clara, CA, USA). This loading process was undertaken in the dark at 37 °C. PBS-loaded discs (8 mm) were used as control. PCAN-loaded discs and the control discs were placed on TSA plates spread with an overnight culture of *S. aureus* and incubated at 37 °C for 24 h and zone of inhibition (ZOI) were measured. Data are presented as mean values of three experiments (n = 3) and error bars represent standard deviation. 

### 3.5. Measurements

#### 3.5.1. Polarized Optical Microscopy (POM)

The three-dimensional fibre matrices before and after degradation were observed with a polarized optical microscope Zeiss (Opton-Axioplan) equipped with a Nikon Coolpix 4500 color digital camera.

#### 3.5.2. Scanning Electron Microscopy (SEM)

SEM of the three-dimensional fibre matrix before and after degradation was conducted using a Quanta 250 FEG (FEI Company, Fremont, CA, USA) high-resolution environmental scanning electron microscope operated at 5 kV acceleration voltage. The samples without coating were analysed under a low vacuum (80 Pa).

#### 3.5.3. Gel Permeation Chromatography (GPC) Analysis

The GPC experiments for the three-dimensional fibre matrix after 1, 15, 31, 51, and 71 days of incubation were conducted in chloroform solution at 35 °C and a flow rate of 1 mL/min using a VE 1122 solvent delivery system (Viscotek, Malvern, UK) with two Mixed C PL-gel styragel columns in series and a Shodex SE 61 refractive index detector (Showa Denko, Munich, Germany). A volume of 10 μL of sample solutions in CHCl_3_ (concentration 0.5% *m*/*v*) were injected into the system. Polystyrene standards with low dispersity were used to generate a calibration curve.

#### 3.5.4. Nuclear Magnetic Resonance ^1^H-NMR Spectroscopy

^1^H-NMR spectra of PLGA, (R,S)-PHB and three-dimensional fibre matrix before and remaining after 1, 15, 31, 51, 65, and 71 days of incubation were recorded using a Bruker-Advance spectrometer operating at 600 MHz (Bruker BioSpin GmbH, Rheinstetten, Germany) with Bruker TOPSPIN 2.0 software using CDCl_3_ as the solvent and tetramethylsilane (TMS) as the internal standard. Spectra were obtained with 64 scans, a 11 μs pulse width, and a 2.66 s acquisition time. 

#### 3.5.5. DSC Analysis

DSC analysis of PLGA, (R,S)-PHB and three-dimensional fibre matrix before and remaining after 71 days of incubation were conducted by means of the TA-DSC Q2000 apparatus (TA Instruments, Newcastle, DE, USA. The calorimetric trace (second heating run) was acquired from −50 °C to 200 °C at a heating rate of 20 °C min^−1^ under nitrogen atmosphere (flow rate = 50 mL min^−1^) for 10 mg of sample. The instrument was calibrated with high purity indium. The glass transition temperature value (T_g_) was taken as the midpoint of the heat capacity step change observed at the second run.

#### 3.5.6. Electrospray Ionization Mass Spectrometry (ESI-MS^n^) Analysis

ESI-MS analysis was performed using a Finnigan LCQ ion trap mass spectrometer (Thermo Fisher Scientific Inc., San Jose, CA, USA). The degradation media containing degradation products were frozen and lyophilized. Then samples were dissolved in chloroform and PCANs were removed via filtration. The solutions of degradation products were dissolved in a chloroform/methanol (1:1 *v*/*v*) system and were introduced to the ESI source via continuous infusion using the instrument syringe pump at a rate of 5 µL/min. The ESI source of the LCQ was operating at 4.5 kV, and the capillary heater was set to 200 °C. Nitrogen was used as the nebulizing gas. The analyses were performed in negative-ion mode.

#### 3.5.7. Determination of the PCAN Release Profile via Ultraviolet–Visible Spectroscopy (UV–VIS) 

The amount of biologically active substance, i.e., PCANs released into the medium from polymeric mats was measured as a function of time. The quantitative assessment of the released PCANs was conducted using UV–VIS Spectrophotometry (Spectrophotometer Spark 10 M, TECAN, Männedorf, Switzerland). The quantity of PCANs released into water after a specified period was determined based on the calibration curve according to the procedure described previously in [29]. A calibration curve (absorbance as a function of PCAN concentration) was generated based on spectrometric measurements of absorbance at 280 nm for 6 diluted PCAN aqueous solutions with a concentration in the range of 0.0–0.5 mg/mL. The percentage of PCANs that was released from the electrospun nonwoven into water after the specified incubation time was calculated based on the calibration curve and taking into account the quantity of PCANs contained in the original samples that were used for the release tests. 

## 4. Conclusions

Based on the polymeric blend of PLGA copolymer and synthetic poly[(R,S)-3-hydroxybutyrate] a novel cytocompatible and antimicrobial system for controlled release of the bioactive substance from *Pelargonium sidoides* was developed. To maintain an optimized level of release of active substances in the disease-affected periodontal region for a sustained prolonged period, biodegradable polymers were used. The developed system of controlled delivery of PCANs through the systematic degradation of the polymer matrix after periodontitis therapy should be completely resorbed by the body. It can also be expected that this novel system could be, in the future, used to deliver drugs in a controlled manner in oral infectious diseases. The results support the potential application of this system for the long-term delivery of various biologically active substances to tissues at the wound site. 

## Figures and Tables

**Figure 1 ijms-25-00503-f001:**
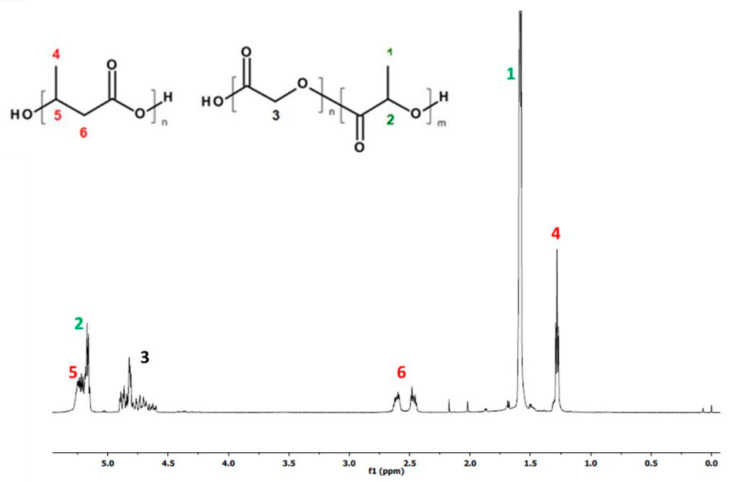
^1^H-NMR spectrum of PLGA/(R,S)-PHB electrospun nonwoven mats obtained.

**Figure 2 ijms-25-00503-f002:**
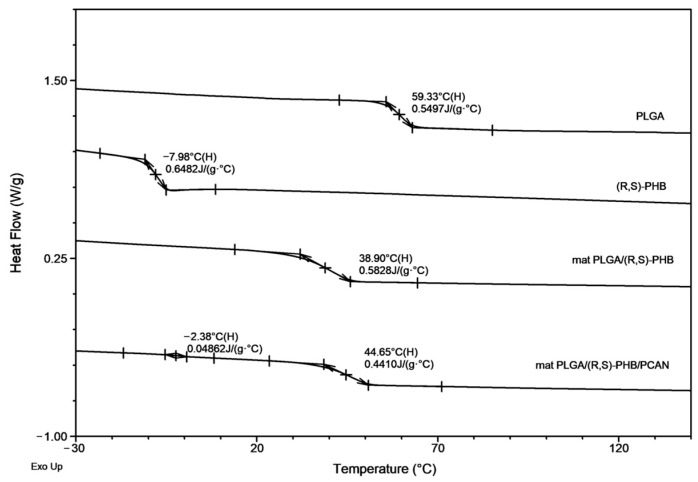
The DSC traces (second heating) for PLGA/(R,S)-PHB electrospun nonwovens with and without PCANs together with traces of PLGA and (R,S)-PHB blend compoents.

**Figure 3 ijms-25-00503-f003:**
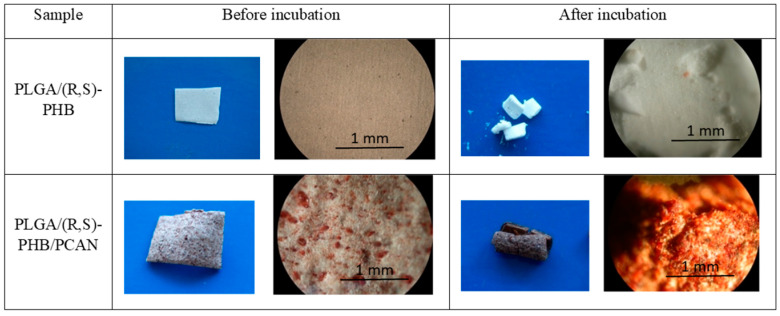
Digital imagines and POM micrographs of PLGA/(R,S)-PHB electrospun nonwoven without and with PCANs before and after 71 days of incubation.

**Figure 4 ijms-25-00503-f004:**
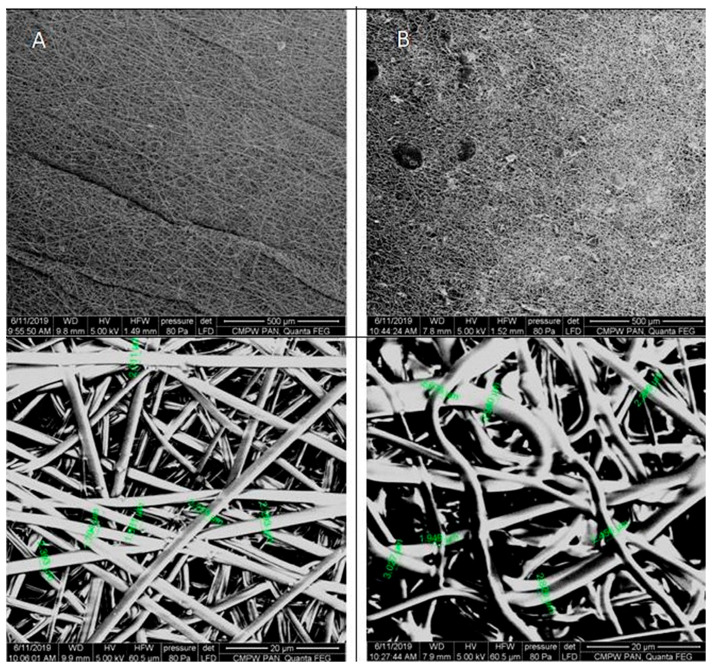
SEM micrographs of PLGA/(R,S)-PHB three-dimensional fibre matrix without (**A**) and with (**B**) a biologically active substance before incubation.

**Figure 5 ijms-25-00503-f005:**
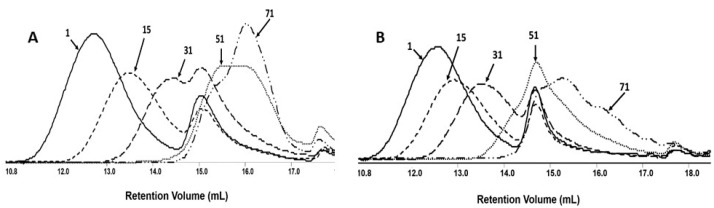
The GPC traces for PLGA/(R,S)-PHB electrospun nonwoven (**A**) without and (**B**) with biologically active substances, i.e., PCANs, during the degradation process.

**Figure 6 ijms-25-00503-f006:**
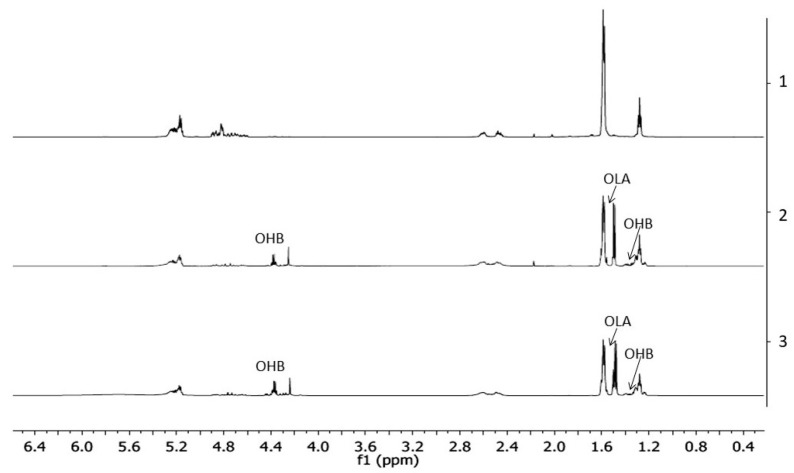
The ^1^H-NMR spectra of the three-dimensional fibre matrix: (1) PLGA/(R,S)-PHB before degradation as well as (2) and (3) PLGA/(R,S)-PHB and PLGA/(R,S)-PHB/PCANs after 71 days of incubation in water, respectively.

**Figure 7 ijms-25-00503-f007:**
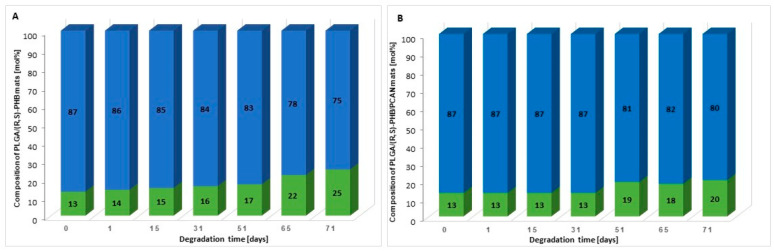
The changes in the chemical composition of (**A**) PLGA/(R,S)-PHB and (**B**) PLGA/(R,S)-PHB/PCAN three-dimensional fibre matrix together with the progress of degradation process estimated based on ^1^H-NMR spectra (blue and green bars represent PLGA and (R,S)-PHB mat components, respectively).

**Figure 8 ijms-25-00503-f008:**
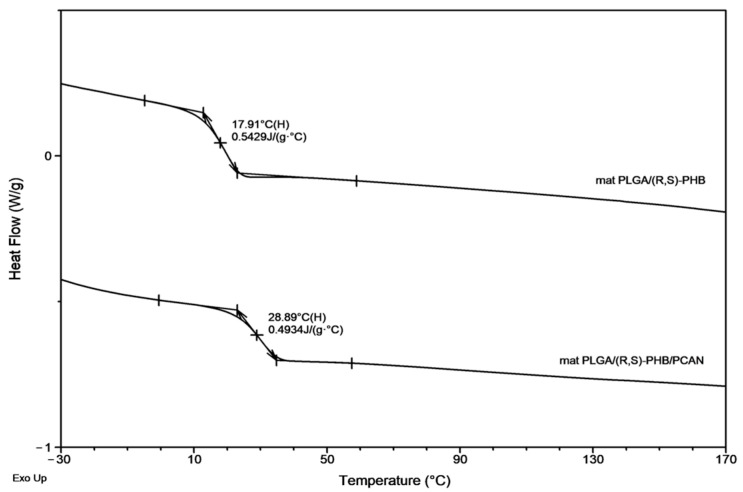
The second DSC heating traces for three-dimensional fibre matrix samples remaining after 71 days of hydrolytic degradation in water.

**Figure 9 ijms-25-00503-f009:**
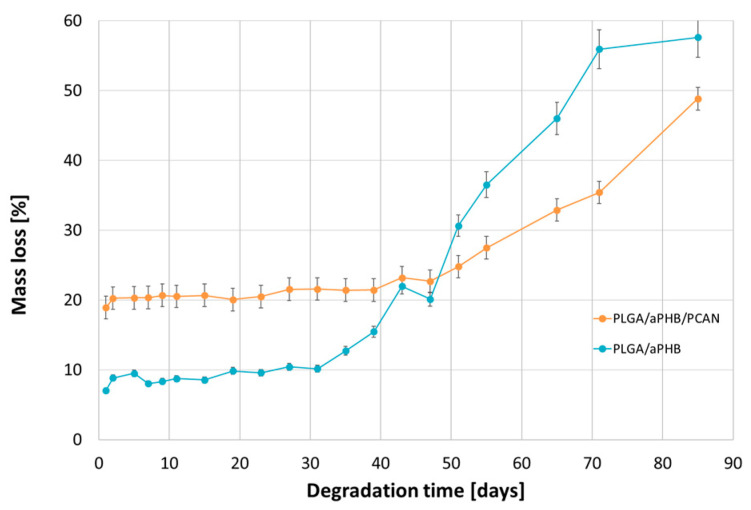
Mass loss of PLGA/(R,S)-PHB and PLGA/(R,S)-PHB/PCAN three-dimensional fibre matrix samples together with the progress of their hydrolytic degradation in water.

**Figure 10 ijms-25-00503-f010:**
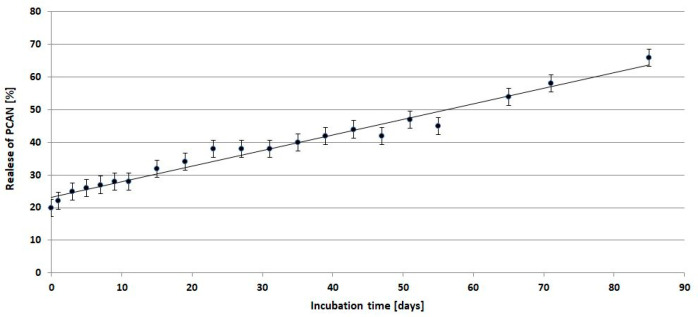
Release profile of PCANs from PLGA/(R,S)-PHB/PCAN electrospun nonwoven during incubation in water.

**Figure 11 ijms-25-00503-f011:**
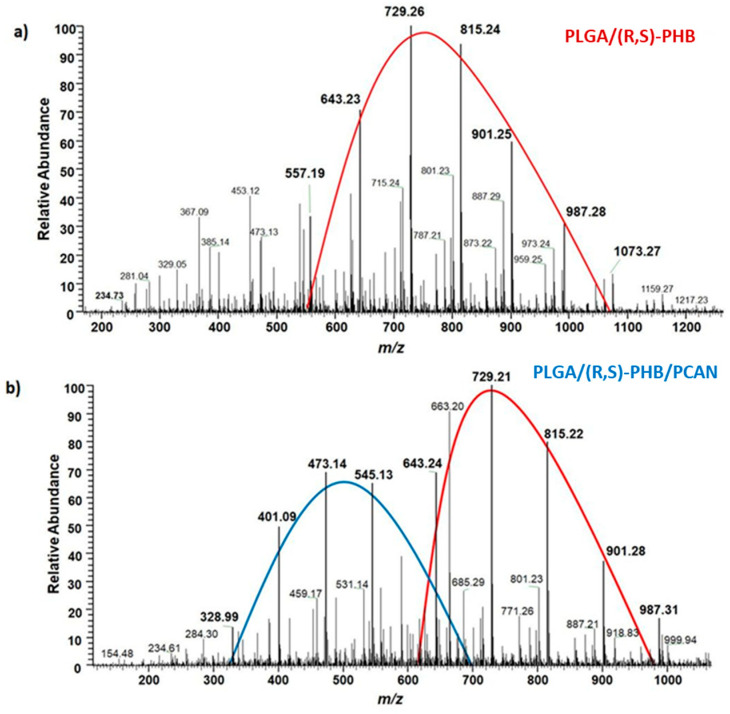
ESI-mass spectra collected after 71 days incubation in water of (**a**) PLGA/(R,S)-PHB and (**b**) PLGA/(R,S)-PHB/PCAN three-dimensional fibre matrix samples.

**Figure 12 ijms-25-00503-f012:**
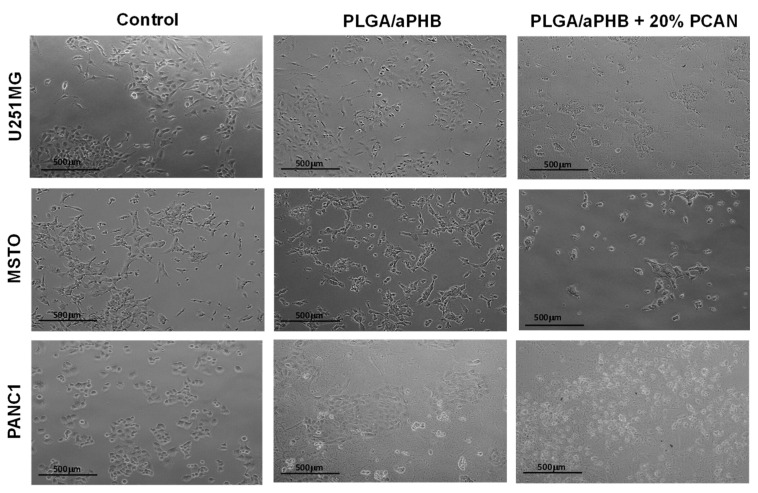
Cytocompatibility test results. Representative optical photomicrographs of cells captured at 10× magnification after 24 h exposure to PLGA/(R,S)-PHB fibre matrix without and with 20 wt% PCANs.

**Figure 13 ijms-25-00503-f013:**
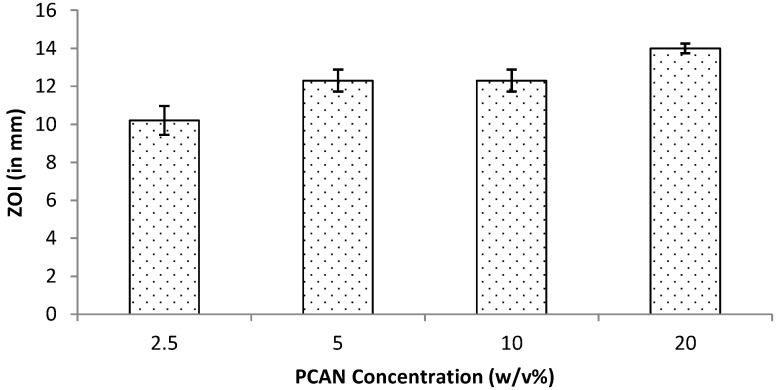
Antimicrobial activity for PCANs against *S. aureus* at 24 h assessed by measuring ZOI during the disc diffusion assay.

**Table 1 ijms-25-00503-t001:** The MTT assay results demonstrating the cell viability of the selected cell lines following a 24 h exposure to media conditioned with PLGA/(R,S)-PHB and PLGA/(R,S)-PHB/PCAN, loaded with 20 wt% of PCANs.

%Cell Viability	PLGA/(R,S)-PHB	PLGA/(R,S)-PHB/PCAN
U251MG ^1^	64.50 ± 1.88%	34.19 ± 3.38%
MSTO ^1^	90.91 ± 6.54%	70.43 ± 3.66%
PANC 1 ^1^	88.62 ± 7.49%	23.73 ± 5.77%

^1^ Cell line.

## Data Availability

Data are contained within the article.

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
