# Peer review of "Designing of Drug Delivery Systems to Improve the Antimicrobial Efficacy in the Periodontal Pocket Based on Biodegradable Polyesters"

_ijms, 2023, doi:10.3390/ijms25010503_

Round 1
Reviewer 1 Report
Comments and Suggestions for Authors
The article ‘Designing of drug delivery systems to improve the antimicrobial efficacy in the periodontal pocket based on biodegradable polyesters’ is well arranged. The article needs to undergo minor revision with the following comments before further proceedings.
1. Abstract: Rewrite the sentence to improve the understanding of the work. ‘The research involved the structural and thermal characteristics of the developed electrospun three-dimensional fiber matrix, as well as hydrolytic degradation tests performed on them’
2. Introduction: The first paragraph of the introduction section is confusing. What do the authors want to discuss, the drug delivery system or the route of administration? The paragraph needs to be properly arranged. The last few lines starting from ‘Moreover,…’ lacks reference.
3. The sentence ‘The aim of this study…’ is very long and advised to fragment it.
4. ‘Pelargonium sidoides (PSRE) is a medical plant…’. Medical plant or medicinal plant.
5. What do authors mean by ‘The antibacterial activities of this plant are mostly assigned to contain proanthocyanidins…’.
6. Line 108: What do authors mean by systematic degradation of polymer matrix?
7. Section 3.1 - looks more like a method rather than materials.
8. Instead of mentioning ‘three-dimensional fiber matrix’ every time, it will be good if provided with some abbreviation.
9. Line 580: Rewrite to provide clarity. ‘The concentrations at [mg/mL]…’.
10. Fig 4: A and B markings are missing.
11. After the addition of PCAN, the authors said that the diameter was changed. What was the final diameter of the matrix?
12. Why did the authors use cancer cell lines for this study when periodontitis is an inflammatory disease? What type of results was expected at the end? How can they justify the results obtained from cancer cell lines that can help this study? If the attempt is made to study the toxicity, then a normal cell line like V79 would have been a better choice. The cell viability reduction denotes the anticancer activity of the formulation, which is completely out of the study plan.
Comments on the Quality of English LanguageSuggested to avoid complex sentences formation.
Author Response
Answers to the comment of Reviewer 1
We would like to thank the Reviewer for the time and effort necessary to review the manuscript. Thank you for any valuable comments and suggestions that helped us improve the quality of our contribution. We have made our best in order to improve the new submission in accordance with the comments. All comments and specific corrections of the Reviewer have duly accepted and introduced into the revised version of the manuscript and have been indicated in yellow colour in manuscript uploaded.
Our position regarding specific Reviewer remarks is presented below:
Comments and Suggestions for Authors
The article ‘Designing of drug delivery systems to improve the antimicrobial efficacy in the periodontal pocket based on biodegradable polyesters’ is well arranged. The article needs to undergo minor revision with the following comments before further proceedings.
- Abstract: Rewrite the sentence to improve the understanding of the work. ‘The research involved the structural and thermal characteristics of the developed electrospun three-dimensional fiber matrix, as well as hydrolytic degradation tests performed on them’
According to the Reviewer 1 suggestion, the indicated part of Abstract was rewritten as follows:
“Delivery systems of the biologically active substances such as proanthocyanidins (PCAN) produced in the form of electrospun nonwoven through the electrospinning method were designed using a polymeric blend of poly(L-lactide-co-glycolide) and poly[(R,S)-3-hydroxybutyrate]. The studies involved the structural and thermal characteristics of the developed electrospun three-dimensional fiber matrixes unloaded and loaded with PCAN . In the next step, the hydrolytic degradation tests of these systems were performed. The release profile of PCAN from the electrospun nonwoven was determined with the aid of UV–VIS spectroscopy.”
- Introduction: The first paragraph of the introduction section is confusing. What do the authors want to discuss, the drug delivery system or the route of administration? The paragraph needs to be properly arranged. The last few lines starting from ‘Moreover,…’ lacks reference.
According to the Reviewer's suggestion, the first paragraph of the Introduction part was rewritten in the revised version of the manuscript together with additional references
Drug delivery devices (DDS) are vehicles for the transportation of therapeutic agents, as well they can maximize the therapeutic effects of the delivered drug while minimizing the undesired side effects [1, 2]. Conventional drug delivery methods are the most popular route for drug administration because they have advantages, such as ease of use and a very high degree of dosage flexibility. However, systemic administration pathways also have disadvantages, such as serious side effects, adverse biodistribution, low selectivity of therapeutic effect, burst release of the drug, and damage to healthy cells [3,4]. DDS in the oral cavity can prepared by various methods to adapt to complex disease situations. Carriers’ materials used in recent years as oral drug delivery sys-tems can be prepared by various methods to adapt to individual patient needs. DDS which can be used in the oral cavity have been classified as nanoparticles, hydrogels, nanofibers, and films [5-7]. Among all these alternatives, electrospinning polymeric fibers (EPFs) produced with biodegradable and biocompatible polymers have proven to be an interesting strategy for drug delivery systems application [5-7].
- The sentence ‘The aim of this study…’ is very long and advised to fragment it.
According to the Reviewer 1 suggestion, the indicated sentence was shortened.
This study aimed to develop a polymeric drug delivery system intended for potential use in periodontitis treatment. The proanthocyanidins, PCAN, extracted from Pelargonium sidoides as an active substance. was used for these studies.
- ‘Pelargonium sidoides (PSRE) is a medical plant…’. Medical plant or medicinal plant.
We thank the reviewer for his valuable remark, the term “ medical plant” has been replaced by “medicinal plant” in the revised version of the manuscript
- What do authors mean by ‘The antibacterial activities of this plant are mostly assigned to contain proanthocyanidins…’.
To clarify the sentence was rewritten in the revised version of the manuscript:
The antibacterial effect of this medicinal plant is mainly attributed to the proanthocyanidins it contains, which also have antioxidant, anti-inflammatory, anti-aging, and anti-cancer properties.
- Line 108: What do authors mean by systematic degradation of polymer matrix?
To clarify this part the sentence was rewritten:
Due to the hydrolytic degradation of the developed electrospun nonwoven fabrics progressing over time, the resulting degradation products should be gradually resorbed by the body and completely absorbed after therapy.
- Section 3.1 - looks more like a method rather than materials.
As a polymeric material for the preparation of the drug delivery system, the blend of poly(L-lactide-co-glycolide) and poly[(R,S)-3-hydroxybutyrate] were used. Both blend components were not commercial products. Therefore, to prepare the delivery system, we were compelled to synthesize these polymers in our laboratory.
Thus, we decided to describe in section 3.1 the methods of the synthesis as well as the characteristics of obtained polymers
- Instead of mentioning ‘three-dimensional fiber matrix’ every time, it will be good if provided with some abbreviation.
To better illustrate the form of the developed systems, the authors would be grateful if the full name of the developed system was retained in the manuscript, not the abbreviation.
- Line 580: Rewrite to provide clarity. ‘The concentrations at [mg/mL]…’.
To clarify the indicated paragraph was revised in the new version of the manuscript as follows.
The amount of PCAN released into the water after a specified period was determined based on the calibration curve according to the procedure described previously in [20]. A calibration curve (absorbance as a function of PCAN concentration) was generated based on spectrometric measurements of absorbance at 280 nm for 6 diluted PCAN aqueous solutions with a concentration in the range of 0.0 - 0.5 mg/ml. The percentage of PCAN that was released from the electrospun nonwoven into the water after the specified incubation time was calculated based on the calibration curve and taking into account the quantity of PCAN contained in the original samples that were used for the release tests.
- Fig 4: A and B markings are missing.
According to Reviewer 1 suggestion, Figure 4 has been corrected in the revised version of the manuscript
- After the addition of PCAN, the authors said that the diameter was changed. What was the final diameter of the matrix?
The fibers in PLGA/(R,S)-PHB electrospun nonwoven studied before degradation possess average diameters of 2.2 µm. The addition of PCAN caused disorganization and entanglement of fibers and influenced varying diameters of single fibers. The average diameters of the fibers after loading of PCAN were range 2.0 –3.0 μm.
However, we did not observe visible changes in the size (diameter) of the electrospun nonwoven.
- Why did the authors use cancer cell lines for this study when periodontitis is an inflammatory disease? What type of results was expected at the end? How can they justify the results obtained from cancer cell lines that can help this study? If the attempt is made to study the toxicity, then a normal cell line like V79 would have been a better choice. The cell viability reduction denotes the anticancer activity of the formulation, which is completely out of the study plan.
Thanks for raising the critical question regarding the selection of cancer cell lines for this study. This paper is a preliminary study focussed on designing of drug delivery systems to improve the treatment in the periodontal pocket. As correctly identified by the reviewer, these studies are not to evaluate anticancer properties of the system. The selected cell lines were purely used as model cells lines to understand the in vitro effect of the novel drug delivery system on the living cells. It is surely beneficial to expand the studies on more relevant cell lines like V79. Since we wanted to get an idea of basic cytocompatibility and any toxicity issues of the developed delivery system, we decided to use the oncogenic cell lines from different tissue sources with different proliferation characteristics which tend to grow quite fast in comparison to normal cells. We infer that if the compound does not affect faster proliferating cells, they will be much safer to cells that proliferate relatively slower such as the normal human fibroblast cell lines.
These studies have given us an initial understanding of the cell lines behaviour with the delivery system. We have reserved further studies on a panel of normal cell lines and fibroblasts for our future study with these hydrogels for a follow up paper.

Reviewer 2 Report
Comments and Suggestions for Authors
Magdalena Zięba et al have presented the research work very well. However, some points need to included before the consideration of this work in IJMS
Specific comments
1. Line 49: specify the examples
2. Line 56: mention the therapeutic modalities used in the treatment of periodontal pocket
3. Line 80 : nightlight the content using reported literature/examples
4. Line 86 : include separate para for the proanthocyanidins mentioning its use in periodontitis, previous report and limitations in treatment
5. Then start how this mat could be potential option in periodontitis
6. Figure 1: provide separate 1H NMR for drug and polymers for better understanding
7. Integration is missing from NMR spectra, required to include in the revised manuscript
8. Figure 2: include the physical mixture in DSC analysis to understand the effect of process on polymer/ drug structure
9. Line 146 : improve the content by providing specific information such as PPM. This part is not convincing
10. Line 153: provide the crucial information from dsc rather than general statement
11. Line 160: instead of visual observation, perform the degradation study on weight basis using suitable time point. Author need to be improve this section in terms of methodology and enrich content by graphical representation
12. Figure 3: provide the scale bar for microscopic images
13. Line 171: How does author confirmed that the bioactive substance distributed uniformly ? specify.
14. Figure 4: label the images properly
15. Line 181: to understand the drug uniformity, authors have advised to perform content uniformity
16. Figure 5: provide the GPC traces for plain drug, drug with PLGA matrix, drug with (R,S)-PHB, and mat containing all polymers for better understanding,
17. Figure 6 : provide the integration
18. Line 214: PLGA undergoes hydrolytic degradation. However, provided information in this para is quite confusing, enrich with proper explanation
19. Figure 7: include the error bar
20. What was the folding endurance of electrospun mat
21. Figure 9: error bars are missing, include in the revised manuscript
22. Figure 10: Include the release profile of plain drug, drug with PLGA mat, drug with (R,S)-PHB and composition of polymers
23. Include reported literature to connect the points in the revised manuscript
24. check for typographical error
Comments on the Quality of English Language
I am not qualified to assess the quality of English in this paper
Author Response
Answers to the comment of Reviewer 2
We would like to thank the Reviewer for the time and effort necessary to review the manuscript. Thank you for any valuable comments and suggestions that helped us improve the quality of our contribution. We have made our best in order to improve the new submission in accordance with the comments. All comments and specific corrections of the Reviewer have duly accepted and introduced into the revised version of the manuscript and have been indicated in blue colour in manuscript uploaded.
Our position regarding specific Reviewer remarks is presented below:
Comments and Suggestions for Authors
Magdalena Zięba et al have presented the research work very well. However, some points need to included before the consideration of this work in IJMS
Specific comments
- Line 49: specify the examples
The indicated paragraph was rewritten, and the examples have been specified in the revised version of the manuscript.
- Line 56: mention the therapeutic modalities used in the treatment of periodontal pocket
According to the Reviewer's suggestion, the paragraph has been rewritten and supplemented as follows:
“Surgical intervention, mechanical therapy, and the use of pharmacological agents are among the most popular approaches used for the treatment of periodontitis disease [10,11]”
- Line 80 : nightlight the content using reported literature/examples
According to the Reviewer's suggestion, the indicated paragraph has been supplemented and additional literature has been added.
“Periodontal treatment involves the preparation of many oral nanofibrous mats that have antibacterial, anti-inflammatory, and tissue regeneration properties [5, 26]. Elec-trospun nanofibers are capable of mimicking the natural extracellular matrix ECM while allowing for the promotion of cell adhesion, proliferation, differentiation, and migration Also, nanofibers can direct cell growth, and they favor precisely oriented tissues, such as stratified periodontal fibers. Hence, electrospun drug-loaded nano-fibrous has the potential to be used effectively in periodontal treatment [5,26]“.
- Line 86 : include separate para for the proanthocyanidins mentioning its use in periodontitis, previous report and limitations in treatment
According to the Reviewer's suggestion, the additional paragraph has been added to the revised version of the manuscript.
“Proanthocyanidins (PCANs) are oligomers or polymers of monomeric flavan-3-ols produced as an end-product of the flavonoid biosynthetic pathway [19-21]. PCANs exhibit antioxidant, anti-inflammatory, cytoprotective, and antimicrobial activity. Proanthocyanidins are a unique class of phytonutrients for the prevention and treatment of periodontal diseases due to their high antimicrobial and immunomodulatory activities. PCANs can also be beneficial in adhesive dentistry due to their natural den-tin cross-linker activity and inhibition of MMPs [20-21]. Using natural, non-toxic products, can make a significant product in alleviating periodontitis symptoms and preventing the disease from progressing [21].”.
- Then start how this mat could be potential option in periodontitis
According to the Reviewer's suggestion, the additional paragraph has been added to the revised version of the manuscript.
“Periodontal treatment involves the preparation of many oral nanofibrous mats that have antibacterial, anti-inflammatory, and tissue regeneration properties [5, 26]. Elec-trospun nanofibers are capable of mimicking the natural extracellular matrix (ECM) while allowing for the promotion of cell adhesion, proliferation, differentiation, and migration Also, nanofibers can direct cell growth, and they favor precisely oriented tissues, such as stratified periodontal fibers. Hence, electrospun drug-loaded nano-fibrous has the potential to be used effectively in periodontal treatment [5,26]“.
- Figure 1: provide separate 1H NMR for drug and polymers for better understanding
Figure 1 shows a spectrum of an electrospun nonwoven loaded with PCAN dissolved in CDCl3.
Because the developed polymeric delivery system dissolves in chloroform and the bioactive substance loaded into it does not dissolve in this solvent, the signals from PCAN are not visible in the spectrum. We mentioned this in the section describing hydrolytic degradation, lines 224-225 of the original manuscript..
However, for better clarity, in the section describing Fig 1 in the revised version of the manuscript we have included the following sentence:
“In Fig. 1, signals from PCAN were not observed because PCAN does not dissolve in CDCl3”.
- Figure 2: include the physical mixture in DSC analysis to understand the effect of process on polymer/ drug structure
and
- Line 153: provide the crucial information from dsc rather than general statement
In the manuscript the authors intended to describe the influence of PCAN on the compatibility and miscibility of the electrospun nonwoven system studied. The effect of the process formation of the electrospun nonwoven is discussed in the context of the miscibility of the components used for the elaboration drug delivery system.
The results of DSC analyses for PLGA/(R,S)-PHB electrospun nonwovens with and without PCAN together with traces of PLGA and (R,S)-PHB blend components indicate, that electrospun nonwoven without PCAN show only one glass transition temperature, Tg with equal 38.9°C which is located between the Tg values of blend components (see Figure 2). Thus, thermal analysis confirms that the electrospinning process does not disturb the miscibility of the blend components. In the case of the electrospun nonwoven loaded with PCAN in DSC trace, two Tg values were observed which indicates that the addition of PCAN to the PLGA/(R,S)-PHB polymer system disrupted its compatibility.
- Line 146 : improve the content by providing specific information such as PPM. This part is not convincing and 7. Integration is missing from NMR spectra, required to include in the revised manuscript
The indicated paragraph started from the line 146 original version was rewritten as follows:
The 1HNMR spectrum revealed the presence of signals corresponding to protons of repeating lactidyl (LA), glycidyl (GA) and 3-hydroxybutyrate (3-HB) units, characteristic of the mixture components (PLGA and poly[(R,S)-3-hydroxybutyrate]) according to with structure (Fig. 1). The chemical composition of electrospun nonwoven samples before degradation was calculated based on the integral values of signals 4 at 1.28 ppm corresponding to the protons of the methyl group 3-HB of the poly[(R,S)-3-hydroxybutyrate] component and signals 1 at 1.58 ppm and 3 at in the range of 4.5-5 ppm represent the protons of the methyl groups LA and the methylene group GA of the PLGA copolymer, respectively. The molar ratio of LA, GA and 3-HB units in the electrospun nonwoven fabric determined on the basis of the 1H-NMR spectrum was 60, 25 and 15 mol%, respectively.
1H NMR spectra of electrospun nonwoven fabric with integrated proton signals characteristic for polymer components are included below.
Figure 1H-NMR spectrum of PLGA/(R,S)-PHB electrospun nonwoven obtained
- Line 160: instead of visual observation, perform the degradation study on weight basis using suitable time point. Author need to be improve this section in terms of methodology and enrich content by graphical representation
The progress of hydrolytic degradation of the polymer carrier in the form of electrospun nonwoven was monitored not only by visual observation of samples using POM and SEM.
The degradation progress was also monitored by determining the change in the average molar masses of the carrier loaded with PCAN and without PCAN
The results are shown in Fig 5 + their discussion on lines 192-206 of the original manuscript
Moreover, using the 1H NMR technique, it was determined how the chemical composition of the polymer carrier changed with the progress of degradation.
Figs 6 and 7 + corresponding discussion 207-246 of the original manuscript.
The progress of the hydrolytic degradation was also monitored by the mass loss measurements PLGA/(R,S)-PHB and PLGA/(R,S)-PHB/PCAN three-dimensional fiber samples after a specific period of degradation Fig 9.
Additionally, The chemical structure of the low-molar mass oligomeric degradation products which were released from the PLGA/(R,S)-PHB and PLGA/(R,S)-PHB/PCAN three-dimensional fibers matrix into the degradation medium was determined by ESI-mass spectrometry Fig 11.
- Figure 3: provide the scale bar for microscopic images
According to the Reviewer's suggestion, this mistake has been corrected in the revised version of the manuscript
- 13. Line 171: How does author confirmed that the bioactive substance distributed uniformly ? specify.
and 15. Line 181: to understand the drug uniformity, authors have advised to perform content uniformity
Distribution of a bioactive substance in an electrospun nonwoven determined based on observations using POM and SEM microscopy. We agree with the Reviewer that the term "distributed uniformly" is not used precisely .As is shown in SEM micrographs Fig 4B the bioactive substance forms different size agglomerates, which are irregularly intercalated between the fibers.
Therefore according to the Reviewer's suggestion, the respective sentence was rewritten
"The digital images, as well as POM micrographs, show that the surface of PLGA/(R,S)-PHB mats before incubation was smooth, and the bioactive substance was distributed in the polymer matrix (Figure 3)."
- Figure 4: label the images properly
According to the Reviewer's requirement,, this mistake has been corrected in the revised version of the manuscript
- Figure 5: provide the GPC traces for plain drug, drug with PLGA matrix, drug with (R,S)-PHB, and mat containing all polymers for better understanding,
Changes in the number-average molar mass with the progress of hydrolytic degradation both for samples of PLGA/(R,S)-PHB electrospun nonwoven without PCAN Fig. 5A and for samples of PLGA/(R,S)-PHB electrospun nonwoven with biological loading active substance PCAN Fig. 5(B) was determined using the GPC method with the aid of chloroform as a solvent of samples and the mobile phase.
The lack of a common solvent for the polymer carrier and the bioactive substance caused difficulties in selecting the mobile phase for GPC analysis. The developed polymeric delivery system dissolves in chloroform and the bioactive substance loaded into it does not dissolve in this solvent,)
To determine changes in molar masses we decided to perform GPC analysis in chloroform. The samples of PLGA/(R,S)-PHB electrospun nonwoven without PCAN dissolved in this solvent very well. In the case of PLGA/(R,S)-PHB electrospun nonwoven samples with biological loading active substance PCAN, the samples were dissolved in chloroform and the insoluble PCAN was before GPC analysis removed by filtration.
Due to the above reasons, and the fact that PCAN does not hydrolytically degrade and only affects the degradation rate of the polymer carrier the GPC analysis of the bioactive substance in chloroform was not performed.
- Figure 6 : provide the integration
As the hydrolytic degradation progresses, due to the formation of low-molecular-weight degradation products, which partly remain occluded in the electrospun nonwoven fabric and partly are resorbed into the degradation medium, the 1HNMR spectra presented in Fig. 6 become more complex.
Therefore, in the manuscript for better readability of Figure 6, the authors decided to present the 1H NMR spectra only with the assignment of the signals which represent the polymer components used without their integration.
Whereas the authors based on the integration of the signals present on 1H-NMR ( recorded before and after specific degradation time) estimated changes in the chemical composition of elaborated electrospun nonwovens PLGA/(R,S)-PHB and PLGA/(R,S )-PHB/PCAN and the results are presented in Figure 7. Below we attach the mentioned spectra for the Reviewer's attention
The 1H-NMR spectra with integration of the three-dimensional fiber matrix: (1) PLGA/(R,S)-PHB before degradation as well as (2) and (3) PLGA/(R,S)-PHB and PLGA/(R,S)-PHB/PCAN after 71 days of incubation in water, respectively.
- Line 214: PLGA undergoes hydrolytic degradation. However, provided information in this para is quite confusing, enrich with proper explanation
To clarify the paragraph indicated by the Reviewer has been rewritten in the revised version of the manuscript as follows:
“The resulting 1H-NMR spectra show the signals of protons characteristic for the LA, GA, and 3-HB repeating units of the polymeric components of electrospun nonwoven accordingly to the structure presented in Figure 1. In addition, in the spectra obtained after 71 days of degradation, proton signals were detected, characteristic of low-molecular-mass 3-hydroxybutyrate (OHB), and OLAGA cooligoesters terminated with hydroxyl and carboxyl end groups. These oligomers are formed during hydrolyses of the electrospun nonwoven components the (R,S)-PHB and PLGA, respectively [26]. The presence of OHB and OLAGA (co)oligomers proton signals with a simultaneous decrease of the intensity of the signals characteristic for repeating units LA, GA, and 3-HB of polymers indicates the progress of the hydrolysis process. Random cleavage of the polymer chains leads to a decrease in the molar mass of the components of the blend. This was confirmed by the GPC results (see Figure 5)
- What was the folding endurance of electrospun mat
Unfortunately, we did not investigate the mechanical properties of the developed electrospun nonwoven drug delivery system.
However, studies on hydrolytic degradation and release of biologically active substances have shown that samples can be manipulated without changing their shape or causing problems with their destruction.
We have reserved mechanical property testing for future research on these systems in a follow-up article
- Figure 9: error bars are missing, include in the revised manuscript
Has been corrected
- Figure 10: Include the release profile of plain drug, drug with PLGA mat, drug with (R,S)-PHB and composition of polymers
Figure 10. shows the release profile of PCAN from PLGA/(R,S)-PHB/PCAN electrospun nonwoven during incubation in water. We agree with the Reviewer that it would be good to compare this profile with the release profile of plain PCAN on a chart. However, the drug PCAN used in this study, when added to water, immediately dissolves completely in it, so we found it unnecessary to present its behaviour on a graph
.23. Include reported literature to connect the points in the revised manuscript
According to the reviewer's suggestion, the manuscript was supplemented with additional literature, and the list of cited literature was renumbered accordingly.
- check for typographical error
According to the Reviewer's suggestion, the text of the manuscript has been checked by a native speaker.

Round 2
Reviewer 2 Report
Comments and Suggestions for Authors
The authors have addressed all comments